environmental science

polyoxometalate, dye decolorization, heteropolyoxotantalate

**Author for correspondence:**
Zhimin Zhou
e-mail: pom@henu.edu.cn

This article has been edited by the Royal Society of Chemistry, including the commissioning, peer review process and editorial aspects up to the point of acceptance.

# Photocatalytic decolorization of three commercial dyes using a new heteropolyoxotantalate catalyst

## Yansong Wang[1,2] and Zhimin Zhou[1]

[1]Institute of Natural Resources and Environment, College of Environment and Planning, and
[2]National Demonstration Center for Environmental and Planning, Henan University, Kaifeng, Henan 475004, People's Republic of China

YW, 0000-0002-3215-8465; ZZ, 0000-0003-4038-3256

The decolorization of commercial dyes is still a pertinent issue since these azo dyes are relatively resistant to conventional biological treatment methods. It is well known that polyoxometalates can absorb light in UV–Vis range that delivers electrons to the reducible species resulting in the decomposition of organic compounds. In this paper, we present the third heteropolyoxotantalate under conventional synthetic conditions in the presence of hydrogen peroxide. The compound has been thoroughly characterized by single-crystal X-ray diffraction, elemental analysis, IR spectroscopy, thermogravimetric analysis and powder X-ray diffraction. The polyanion incorporates two 3-peroxotantalo-2-phosphate clusters that are linked together by four oxygen bridges. In addition, the photocatalytic activities of the title compound **1a** were investigated. After 270 min irradiation, about 90% of Rhodamine B (RhB) was removed in the presence of **1a** while the degradation of RhB could be negligible in the absence of **1a**, indicating it can be a promising catalyst candidate for decolorization of organic dyes. Also, photocatalytic experiment for hydrogen generation was studied, and the results show that the $H_2$ evolution rate is 3383 µmol h$^{-1}$ g$^{-1}$ for compound **1a** (100 mg) over 6 h with the corresponding turnover number of 432.

## 1. Introduction

Nowadays, azo dyes are widely used in various industrial areas such as textiles, cosmetics, ceramics, leather, paper and food processing [1]. It is estimated that over 0.7 million tons of

synthetic dyes are annually manufactured [2], and they are significantly lost to effluents by 5–15% amount during dyeing and finishing operations as a result of inefficiency in the dyeing process [3]. Most of these waste dyes in the environment are stable against temperature, light, detergents and microbial attack [4]. Further, they are believed to be toxic and non-biodegradable in nature, which inevitably poses many severe hazards on both human health and ecological systems [1,5]. Therefore, it is urgent to develop methods for treatment of organic dyes. Among them, the economical and effective photocatalytic method has attracted more attention [6–9].

Polyoxometalates (POMs) [10] are a unique family of polynuclear anionic metal oxo clusters with properties suitable for many potential applications in catalysis, magnetism, biomedicine, materials science and nanotechnology [11,12]. Concerning the first topic, the use of POM clusters as catalysts continues to be the most popular in this field. Up to now, a large number of POMs with interesting properties have been reported [13,14]. In particular, the number of publications concerning hetero-POMs over the last two decades has largely arisen as a result of the use of lacunary heteropolyoxoanions, which function as multidentate ligands to bind other metal ions, giving a plethora of new species [15]. However, the V group of Nb and Ta are significantly different from the well-known VI group of W, Mo or V-based POMs. The latter can easily self-assemble to polynuclear clusters via acidification of aqueous monomeric oxoanions. Generally, Nb and Ta are expected to present similar behaviour, the research on polyoxoniobates increases exponentially since the intriguing cluster reported in 2002 [16]. However, polyoxotantalate (POTa) chemistry has been far less investigated than that of niobium analogue, although the single-crystal X-ray study on $K_7H[Nb_6O_{19}] \cdot 13H_2O$ and $K_8[Ta_6O_{19}] \cdot 16H_2O$ was reported as early as 1953 [17] and 1954 [18], respectively.

Over the past few decades, there have been scattered reports on POTa compounds. The scanty development and interest is mainly focused on iso-POTas (IPOTas). In 1963, Nelson & Tobias first carried out an investigation [19] which indicated that the hexatantalate anion ($[Ta_6O_{19}]^{8-}$, $Ta_6$) in the crystal also exists in aqueous solution. Recently, Nyman and co-workers have directly observed the ion-association behaviour of aqueous $Ta_6$ by using small-angle X-ray scattering [20]. Meanwhile, the preparation of this and related salts has been studied by several other workers [21–23]. In 2011, Hu and co-workers [24] communicated two novel POTa derivatives which are constructed from the Lindqvist-type hexatantalate anion and copper–amine complexes. In 2012, Yagasaki and co-workers [23] reported a novel hexatantalate tetramer in which four $Ta_6$ units are connected by 18 hydrogen bonds to form a rod-shaped supramolecule. However, decatantalate has been isolated as a tetrabutylammonium salt from non-aqueous solution until 2013 [25]. Recently, Liu *et al.* [26] and Huang *et al.* [27] reported several Ta/W mixed-addendum POMs, respectively. In 2016, Son & Casey [28] communicated two Ti-substituted POTa clusters, $[Ti_2Ta_8O_{28}]^{8-}$ and $[Ti_{12}Ta_6O_{44}]^{10-}$. Very recently, Niu and co-workers [29] reported two 6-Peroxotantalo-4-phosphate clusters, from the viewpoint of structure, which can be regarded as the first two examples of hetero-POTas (HPOTas).

Herein, we present the synthesis, structure and photocatalytic properties of a new HPOTa complex $K_3[H_3P_4(TaO_2)_6(OH)_4O_{20}] \cdot 12H_2O$ (K$_3$-**1**-12H$_2$O, **1a**). To the best of our knowledge, compound **1a** represents the third example of HPOTa, but it is the first time to report the photocatalytic properties.

# 2. Experimental section

## 2.1. Material and methods

All reagents and solvents were obtained from commercial suppliers and used as received. $K_8[Ta_6O_{19}] \cdot 16H_2O$ was prepared using literature methods [19]. The IR spectra (using KBr in pellets) were recorded on a Bruker VERTEX 70 IR spectrometer (4000–450 cm$^{-1}$). X-ray powder diffraction spectral data were recorded on a Bruker AXS D8 Advance diffractometer with Cu K$\alpha$ radiation in the angular range $2\theta = 5$–45° at 293 K. K, P and Ta elemental analyses were obtained with a PerkinElmer Optima 2100 DV inductively coupled plasma optical emission spectrometer. UV–Vis spectra were obtained with a U-4100 spectrometer at room temperature.

## 2.2. Synthesis of **1a**

$K_8[Ta_6O_{19}] \cdot 16H_2O$ (0.3 g, 0.15 mmol) was dissolved in a solution consisting of 2.7 ml of 30% aqueous $H_2O_2$ and 33 ml of water. Diluted phosphoric acid (3 mol l$^{-1}$, 1.3 ml) was added dropwise under rapid stirring for 15 min, resulting in a clear solution. The pH of the resulting mixture was adjusted to

**Table 1.** Crystal data and structure refinement of compound **1a**.

| | 1a |
|---|---|
| formula | $K_3H_{31}P_4Ta_6O_{48}$ |
| $M_r$ (g mol$^{-1}$) | 2126.09 |
| $T$ (K) | 296.15 |
| crystal system | monoclinic |
| space group | $P2_1/c$ |
| $a$ (Å) | 12.6862 (10) |
| $b$ (Å) | 9.9322 (8) |
| $c$ (Å) | 16.8359 (13) |
| $\beta$ (°) | 105.3940 (10) |
| volume (Å$^3$) | 2045.2 (3) |
| $Z$ | 2 |
| $D_c$ (g cm$^{-3}$) | 3.441 |
| $\mu$ (mm$^{-1}$) | 16.581 |
| crystal size (mm$^3$) | $0.41 \times 0.25 \times 0.09$ |
| limiting indices | $-10 \leq h \leq 15$ |
| | $-11 \leq k \leq 11$ |
| | $-20 \leq l \leq 16$ |
| reflns collected | 10 225 |
| indep reflns | 3624 |
| $R_{int}$ | 0.0321 |
| GOF on $F^2$ | 1.062 |
| $R_1$[a], $wR_2[I > 2\sigma(I)]$[b] | 0.0230, 0.0540 |
| $R$ indices (all data) | 0.0266, 0.0554 |

[a]$R_1 = \sum ||F_o| - |F_c||/\sum |F_o|$.

[b]$wR_2 = \left\{ \sum [w(F_o^2 - F_c^2)^2]/\sum [w(F_o^2)^2] \right\}^{1/2}$.

2.8 by 2 mol l$^{-1}$ KOH aq and then heated to 90°C for 3 h. After this period, the mixture was cooled to room temperature and filtered, followed by the addition of KCl (0.12 g, 1.6 mmol). The solution was then stirred for half an hour and filtered. The resulting filtrate was kept at room temperature to allow slow evaporation for about one week (yield 0.18 g, 58% based on Ta). IR (KBr, cm$^{-1}$): 1159, 1081, 1011, 955, 852, 840, 797, 674, 583 and 532 cm$^{-1}$; analysis (calcd, found for $K_3H_{31}O_{48}P_4Ta_6$): K (5.52, 5.57), P (5.83, 5.86), Ta (51.1, 50.7).

## 2.3. X-ray crystal-structure analyses

Suitable single crystals were selected from their respective mother liquors and placed in a thin glass tube. X-ray diffraction intensity was recorded on a Bruker Apex-II CCD diffractometer at 296 (2) K with MoKa monochromated radiation ($\lambda = 0.71073$ Å). Structure solution and refinement were carried out by using the SHELXS-97 and SHELXL-2014 program packages [30,31] for **1a**. CCDC 1573496 for **1a** contains the electronic supplementary material, crystallographic data [32]. These data are provided free of charge by The Cambridge Crystallographic Data Centre. Selected details of the data collection and structural refinement of compound **1a** can be found in table 1.

## 2.4. Dye decolorization

The photocatalytic activity of the title compound was demonstrated by studying the change of the absorbance intensity of Rhodamine B (RhB), Methyl blue (MB) and Acid red 1 (AR1). They were

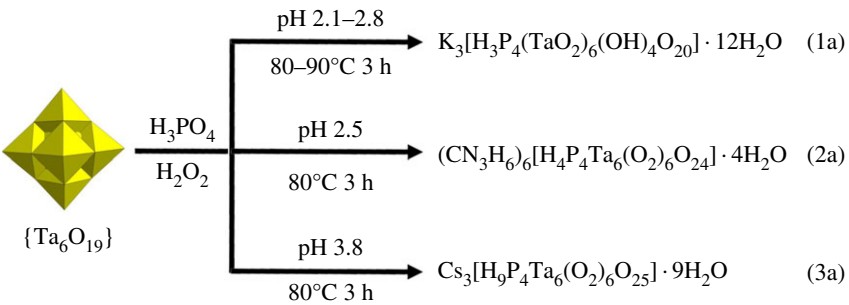

$$\{Ta_6O_{19}\} \xrightarrow{\substack{H_3PO_4 \\ H_2O_2}} \begin{cases} \xrightarrow[80–90°C\ 3\ h]{pH\ 2.1–2.8} K_3[H_3P_4(TaO_2)_6(OH)_4O_{20}] \cdot 12H_2O \quad (1a) \\ \xrightarrow[80°C\ 3\ h]{pH\ 2.5} (CN_3H_6)_6[H_4P_4Ta_6(O_2)_6O_{24}] \cdot 4H_2O \quad (2a) \\ \xrightarrow[80°C\ 3\ h]{pH\ 3.8} Cs_3[H_9P_4Ta_6(O_2)_6O_{25}] \cdot 9H_2O \quad (3a) \end{cases}$$

**Scheme 1.** Synthetic procedures leading to the isolation of compounds **1a**, **2a** [29] and **3a** [29], highlighting the effects of pH and cation.

dissolved in water (35 mg l$^{-1}$) and the dye decolorization experiments were performed in an open batch system. In a typical run, 2 ml of dye solution, 40 ml water and a certain amount of **1a** were mixed and reacted in ambient conditions under the irradiation of the 350 W Xenon lamp with magnetic stirring. The decolorization rate of RhB was evaluated using the UV–vis absorption spectra to measure the peak value of a maximum absorption of RhB solution (554 nm). During the irradiation, 4 ml of mixture solution was pipetted into a quartz cell at given time intervals and measured by a U-4100 spectrometer in the range of 400–700 nm at room temperature. The reaction was carried out at 298 K and all catalysis tests were analysed in triplicate.

# 3. Results and discussion

## 3.1. Synthesis

Compound **1a** was obtained by a simple one-pot reaction of potassium hexatantalate with phosphoric acid in the presence of $H_2O_2$. The solution was adjusted to pH 2.8 and heated to 90°C for 3 h, followed by the addition of KCl. Interestingly, the synthetic procedure for **1a** is similar to that for $(CN_3H_6)_6[H_4P_4Ta_6(O_2)_6O_{24}] \cdot 4H_2O$ (**2a**) and $Cs_3[H_9P_4Ta_6(O_2)_6O_{25}] \cdot 9H_2O$ (**3a**) reported very recently [29]. We found that the key factors determining whether **1a**, **2a** or **3a** is formed appear to be pH and cation. This work demonstrates that tiny changes in the synthetic conditions may have huge impact on the product formed. As shown in scheme 1, **1a** and **2a** can be obtained when the solution was adjusted to pH 2.5 and then heated to 80°C for 3 h, followed by the addition of potassium and guanidinium ion, respectively. On the other hand, compound **3a** can be obtained if the pH was adjusted to 3.8 as well as the need for caesium ions. In addition, compound **1a** can be also synthesized in the range of pH 2.1–2.8 and temperature 80–90°C.

## 3.2. Structural analysis

Single-crystal X-ray diffraction analysis reveals that compound **1a** crystallizes in the monoclinic space group $P2_1/c$ and comprises a $[H_3P_4(TaO_2)_6(OH)_4O_{20}]^{3-}$ (**1**) polyanion, three potassium count cations and 12 water molecules. The crystal structure of **1** resembles that of a previously reported cluster $[H_4P_4Ta_6(O_2)_6O_{24}]^{6-}$, in which two 3-peroxotantalo-2-phosphate $\{P_2Ta_3\}$ (electronic supplementary material, figure S1a) fragments are fused together via four bridging oxygen atoms (M–O–Ta, M = Ta/P), resulting in a *trans*-condensed cluster. To our knowledge, cluster **1** represents the third example of heteropolyoxotantalate (figure 1; electronic supplementary material, figure S2). Each of the six Ta atoms is coordinated by five oxygen atoms and one peroxo group, resulting in a distorted pentagonal–bipyramidal coordination geometry, whereas all the P atoms exhibit conventional tetrahedral coordination polyhedra (electronic supplementary material, figure S1b,c). In **1**, the Ta–O and P–O bond lengths are in the range of 1.916 (4)–2.099 (4) and 1.485 (5)–1.559 (5) Å, respectively. Interestingly, different from the reported peroxotantalum-substituted POMs [26], the average value of the $O_P - O_P$ bond in **1** (1.494 Å) is almost identical to that for non-coordinated $O_2^{2-}$ (1.49 Å) [33]. Alternatively, the structure of polyanion **1** is similar to that of $P_4M_6$ cluster (M = Nb/Ta) [29], with four phosphate ligands stabilizing the peroxo-$\{M_6\}$ cluster. As expected, the metal–oxygen bond lengths in **1** compare well to those in the previously isolated $P_4M_6$ cluster (M = Nb/Ta; electronic supplementary material, table S1).

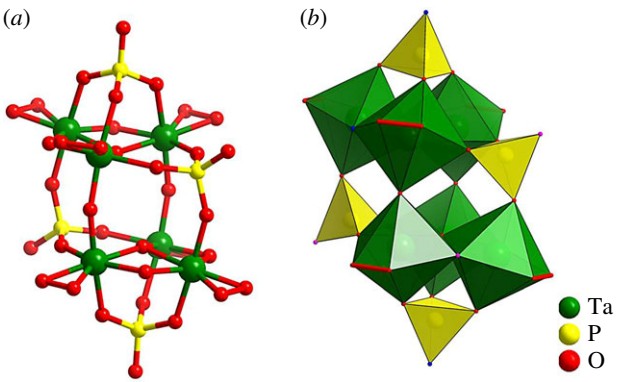

**Figure 1.** Ball-and-stick (*a*) and polyhedral (*b*) representations of polyanion **1**. All cations and solvent water molecules have been omitted for clarity.

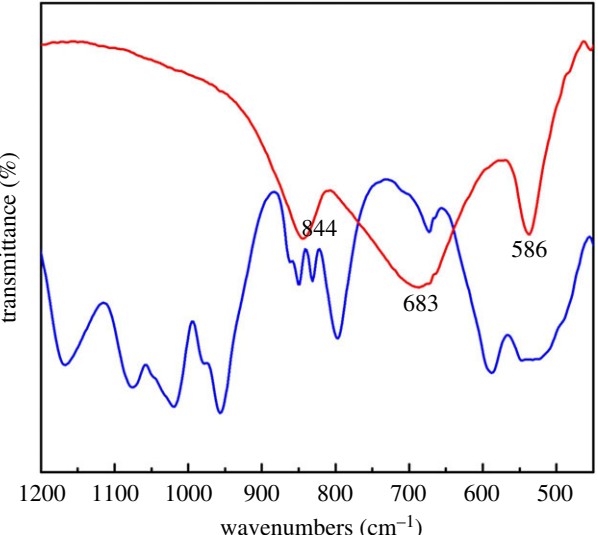

**Figure 2.** IR spectra of **1a** (blue) and $K_8[Ta_6O_{19}] \cdot 16H_2O$ (red) in the region between 1200 and 450 $cm^{-1}$.

The oxidation state of the phosphorus and tantalum centres was confirmed by bond valence sum (BVS) calculations [34] (electronic supplementary material, table S2). Also, the results from X-ray crystal structure determination and element analysis required seven additional protons for charge balance. The P2–O18 bond length in **1** is 1.50 Å, and BVS calculations suggest that these two O18 terminal oxygen atoms (shown in pink in electronic supplementary material, figure S3) have one protons associated with them (P–OH). In addition, the BVS of the O10 atom bridging Ta2 and Ta3 centres is 1.32, indicating mono-protonated (shown in pink in electronic supplementary material, figure S3). Meanwhile, the intermediate BVS value of 1.38 shows that the two terminal O16 atoms and the two bridging O4 atoms are occupied by O/OH ligand with an occupancy factor of 0.25 for O (shown in blue in electronic supplementary material, figure S3). Thus, polyanion **1** should be described as $[H_3P_4(TaO_2)_6(OH)_4O_{20}]^{3-}$.

## 3.3. IR spectra

The Fourier transform infrared spectra (FTIR) of compounds **1a** and $K_8[Ta_6O_{19}] \cdot 16H_2O$ are shown in figure 2; electronic supplementary material, figure S4. The IR spectrum of **1a** displays several strong and medium bands in the range of 1200–1000 $cm^{-1}$, associated with antisymmetric stretching of the P–O bond. As shown in figure 2, the Ta=O band at 840 $cm^{-1}$ and the Ta–O–Ta band at 674 and 583 $cm^{-1}$ in **1a** are at similar positions in $K_8[Ta_6O_{19}] \cdot 16H_2O$. However, the bands at 797 and 583 $cm^{-1}$ are much more pronounced in **1a** than in $K_8[Ta_6O_{19}] \cdot 16H_2O$, which may be assigned to P–O–Ta vibration modes. Compared with that of the precursor $K_8[Ta_6O_{19}] \cdot 16H_2O$, the significant changes in FTIR spectrum of **1a**

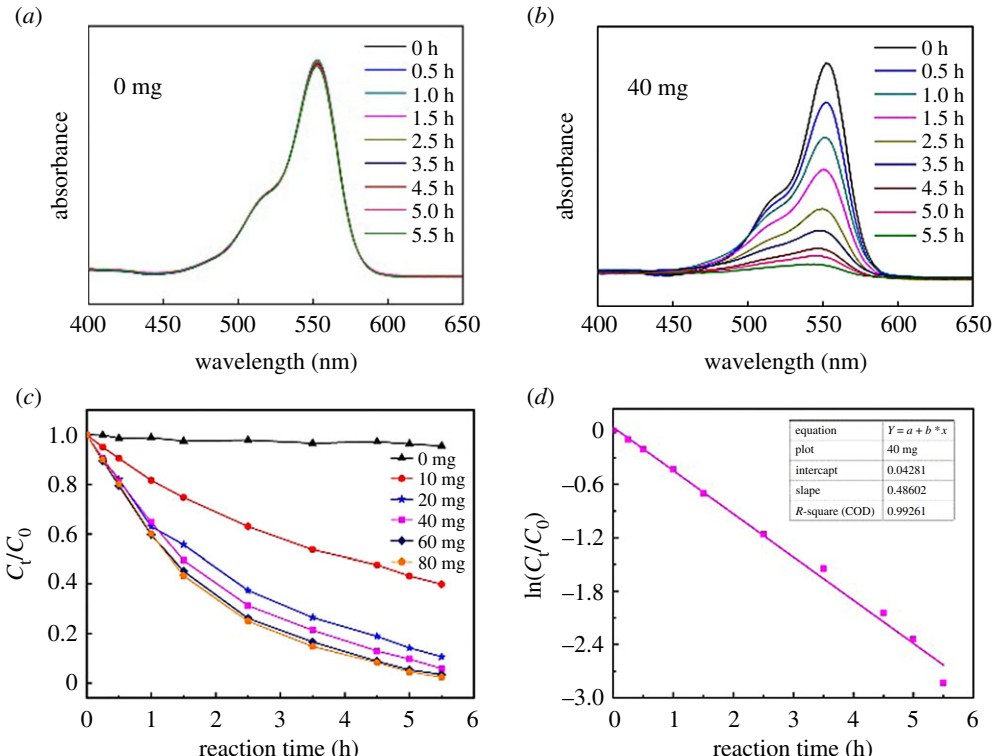

**Figure 3.** Photocatalytic performance of RhB. (*a*) The blank experiment in the absence of **1a**; (*b*) absorbance as a function of time of RhB using **1a** (40 mg); (*c*) plot of $C_t/C_0$ versus time with different amounts of **1a**.; and (*d*) the first-order linear plot of $\ln(C_t/C_0)$ versus time for RhB.

are the appearance of strong intensity peaks in the region 1200–1000 cm$^{-1}$ and medium intensity band at 852 cm$^{-1}$ (figure 2), which is characteristic of the antisymmetric stretching vibrations of P–O bond and peroxo group [35], respectively. This is in good agreement with the solid-state structure. In addition, X-ray powder diffraction pattern of compound **1a** agrees well with its simulated pattern based on the single-crystal (electronic supplementary material, figure S5), indicating the phase purity of the materials.

## 3.4. Photocatalytic studies

Catalysis has been the most promising application in POM chemistry and they are widely studied as catalysts in many fields [36]. Synthetic dyes are environmental hazards because they are difficult to decompose by natural means, and degrading textile dyes by photocatalysis has been studied extensively [37–40]. Thus, in this work, we intend to investigate the photocatalytic behaviours for the decolorization of dyes under visible-light irradiation. The catalytic activity of **1a** was investigated by using commercial dyes (RhB, MB and AR1; electronic supplementary material, figure S6). The catalysis reactions were monitored by the decrease in absorbance at $\lambda$max 554, 598 and 530 nm for RhB, MB and AR1 with time, respectively. The catalytic decolorization studies were carried out in the absence and presence of **1a**. As shown in figure 3*a* and electronic supplementary material, figure S7, the blank experiments conducted without **1a** or lamp showed almost no change in colour as well as the intensity of $\lambda$max at 554 nm in the case of RhB. But the rate of decolorization was greatly enhanced upon the addition of even a small amount of the catalyst indicating the immense catalytic effect of **1a** in this reaction. This was evident from the bleaching of the red colour of RhB as well as the decrease in the intensity of $\lambda$max. For comparison, the catalytic performances of different amounts of **1a** (10, 20, 40, 60 and 80 mg) on the decolorization of RhB dyes were also investigated. It can be seen that when the amount of **1a** is more than or equal to 40 mg, the RhB was completely decolorized within 5.5 h (figure 3*b*; electronic supplementary material, figure S8).

The catalytic reaction could be considered as a pseudo-first-order kinetics with regard to the linear fit of the $\ln(C_t/C_0)$ data. The rate constant ($k_{app}$, h$^{-1}$) was determined from the following rate equations:

$$\frac{\mathrm{d}C_t}{\mathrm{d}t} = -k_{app} \times C_t \tag{3.1}$$

and

$$\ln(C_t|C_0) = \ln(A_t|A_0) = -k_{\text{app}} \times t, \tag{3.2}$$

where $C_t$ represents the concentration of dye, $t$ is the reaction time, $A_0$ and $A_t$ are the absorbance of RhB (554 nm) at time 0 and $t$, respectively. Therefore, as shown in figure 3$d$, the calculated $k_{\text{app}}$ for the reduction in RhB in the presence of **1a** is $0.487\,\text{h}^{-1}$. Moreover, total organic carbon (TOC) concentration of RhB solution treated by **1a** is analysed, which achieved total TOC mineralization of 56% (electronic supplementary material, figure S9). In addition, the stability of compound **1a** in solution can be proved by UV (electronic supplementary material, figure S10) and ESI-MS spectra (electronic supplementary material, figure S11) spectra, while the comparison of IR spectra before and after catalysis (electronic supplementary material, figure S12) indicates the stability of compound **1a** in solid state.

Photocatalytic reaction on **2a** was also investigated for comparison to **1a**, which shows the similar catalytic properties for the decolorization of RhB dye (electronic supplementary material, figure S13). This may be attributed to the fact that **1a** is a structural analogue of **2a**. Moreover, compound **1a** also exhibits photocatalytic activities for the decolorization of MB (electronic supplementary material, figure S14) and AR1 (electronic supplementary material, figure S15) but with relatively low catalytic performances compared to RhB.

Photocatalytic water-splitting offers a promising way for environmentally friendly solar-hydrogen production in recent years. Thus, a preliminary photocatalytic study for hydrogen generation has been done. The results show that the $H_2$ evolution rate is $3383\,\mu\text{mol}\,\text{h}^{-1}\,\text{g}^{-1}$ for compound **1a** (100 mg) over 6 h with the corresponding turnover number of 432 (moles of $H_2$ formed/moles of **1a**), which is shown in electronic supplementary material, figure S16. Also, blank experiments indicate that no $H_2$ can be detected under the absence of Pt-co-catalyst, sacrificial solvent ($CH_3OH$) or cluster **1a**.

## 4. Conclusion

The third heteropolyoxotantalate cluster known so far, $K_3[H_3P_4(TaO_2)_6(OH)_4O_{20}] \cdot 12H_2O$ (**1a**), has been successfully synthesized. Interestingly, the synthesis of **1a** leads us to believe that the peroxotantalate may be a potential active site to react with the classic hetero atoms, such as 3d transition-metal or lanthanide ions, providing an alternative perspective in POTa chemistry. Moreover, **1a** can be used to degrade three organic dyes under visible condition. It also exhibits photocatalytic $H_2$ evolution activity.

Data accessibility. Electronic supplementary material contains additional structural information (electronic supplementary material, figures S1–3), bond lengths (electronic supplementary material, table S1), BVS calculation results (electronic supplementary material, table S1), IR spectra (electronic supplementary material, figures S4 and S12), XRD patterns (electronic supplementary material, figure S5), photocatalytic performances of three commercial dyes (electronic supplementary material, figures S6–S9, S13–S15), UV spectra (electronic supplementary material, figure S10), ESI-MS spectra (electronic supplementary material, figure S11), photocatalytic $H_2$ evolution (electronic supplementary material, figure S16) and its mechanism illustration (electronic supplementary material, figure S17). See doi:10.6084/m9.figshare.8266796.

Authors' contributions. Y.W. and Z.Z. conceived the project; Y.W. prepared the solid samples, carried out the characterization; Y.W. and Z.Z. analysed the data and wrote the manuscript. All authors gave final approval for publication.

Competing interests. We have no competing interests.

Funding. This work was supported by National Natural Science Foundation of China (41401503), the NSF from Henan Province (162300410132) and the Program for Innovative Research Team (in Science and Technology) in University of Henan Province (6IRTSTHN012). Y.W. acknowledges China Scholarship Council (201607900006) for the financial support.

Acknowledgements. We gratefully thank Dr Junyong Li and Dr Guan Wang for TOC experiment and photocatalytic study for hydrogen generation, respectively.

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
