## [Reviewer comments · Royal Society Open Science]

Review History

RSOS-190015.R0 (Original submission)

Review form: Reviewer 1

Is the manuscript scientifically sound in its present form?

Yes

Are the interpretations and conclusions justified by the results?

Yes

Is the language acceptable?

Yes

Is it clear how to access all supporting data?

Yes

Do you have any ethical concerns with this paper?

No

Have you any concerns about statistical analyses in this paper?

I do not feel qualified to assess the statistics

Recommendation?

Major revision is needed (please make suggestions in comments)

Comments to the Author(s)

This manuscript reported the synthesis of a new heteropolyoxotantalate and tested its performance for photocatalytic degradation of three organic dyes in water. The materials are generally well characterized. However, this reviewer has concerns over the rationale for using this homogeneous photocatalyst for dye removal, since the compound is hard to remove from water after photocatalytic test. Actually, this may be the reason why the compound (and other homogeneous photocatalysts) are more suitable for solar energy applications such as hydrogen generation instead of environmental applications such as water purification. To be accepted for publication, the authors need to address the above concern and performance experiments to show that this compound can be easily recovered from the treated water after photocatalyst tests and possibly reused.

Review form: Reviewer 2**Is the manuscript scientifically sound in its present form?**

Yes

Are the interpretations and conclusions justified by the results?

Yes

Is the language acceptable?

Yes

Is it clear how to access all supporting data?

Yes

Do you have any ethical concerns with this paper?

No

Have you any concerns about statistical analyses in this paper?

No

Recommendation?

Accept with minor revision (please list in comments)

Comments to the Author(s)

In this manuscript, the authors present the synthesis, structure and its photocatalytic properties of a new HPOTa complex $K_3[H_3P_4(TaO_2)_6(OH)_4O_{20}] \cdot 12H_2O$, which was successfully used to degrade the Rhodamine B, Methyl blue and Acid red 1. The characterization and data analysis of this manuscript is sufficient. Therefore, the manuscript can be considered to be accepted after further revision. The following questions need to be addressed for further improvement:

1. The nature of catalyst is that the composition of catalyst does not change before and after the catalytic reaction. In order to prove that the as-obtained material is a catalyst, please provide the phase data of the material after the catalytic reaction.
2. Dye-degradation catalysts have been studied for decades. Many corresponding catalysts have

been developed. Please compare the previous reports on dyestuff degradation catalysts and propose the advantages of using HPO₃Ta complex as a catalyst. The author can refer to relevant research, such as *Nanoscale*, 2013, 5, 6589-6598; *Nanoscale*, 2014, 6, 1124-1133; *CrystEngComm*, 2012, 14, 3965-3971; *Chemical Engineering Journal*, 2018, 334, 1537-1549.

3. Please propose the mechanism of catalytic degradation.

4. There are some typo and grammar mistakes. Please revise the manuscript carefully.

Decision letter (RSOS-190015.R0)

22-May-2019

Dear Dr Zhou:

Title: Photocatalytic decolorization of three commercial dyes using a new heteropolyoxotantalate catalyst

Manuscript ID: RSOS-190015

The editor assigned to your manuscript has now received comments from reviewers. We would like you to revise your paper in accordance with the referee and Subject Editor suggestions which can be found below (not including confidential reports to the Editor). Please note this decision does not guarantee eventual acceptance. I apologise that this took longer than usual.

Please submit your revised paper before 14-Jun-2019. Please note that the revision deadline will expire at 00.00am on this date. If we do not hear from you within this time then it will be assumed that the paper has been withdrawn. In exceptional circumstances, extensions may be possible if agreed with the Editorial Office in advance. We do not allow multiple rounds of revision so we urge you to make every effort to fully address all of the comments at this stage. If deemed necessary by the Editors, your manuscript will be sent back to one or more of the original reviewers for assessment. If the original reviewers are not available we may invite new reviewers.

Please also include the following statements alongside the other end statements. As we cannot publish your manuscript without these end statements included, if you feel that a given heading is not relevant to your paper, please nevertheless include the heading and explicitly state that it is not relevant to your work.

- Acknowledgements

On behalf of the Subject Editor Professor Anthony Stace and the Associate Editor Professor Claire Carmalt.

RSC Associate Editor:
Comments to the Author:
(There are no comments.)

RSC Subject Editor:
Comments to the Author:
(There are no comments.)

Reviewers' Comments to Author:
Reviewer: 1

Comments to the Author(s)

This manuscript reported the synthesis of a new heteropolyoxotantalate and tested its performance for photocatalytic degradation of three organic dyes in water. The materials are generally well characterized. However, this reviewer has concerns over the rationale for using this homogeneous photocatalyst for dye removal, since the compound is hard to remove from water after photocatalytic test. Actually, this may be the reason why the compound (and other homogeneous photocatalysts) are more suitable for solar energy applications such as hydrogen generation instead of environmental applications such as water purification. To be accepted for publication, the authors need to address the above concern and performance experiments to show that this compound can be easily recovered from the treated water after photocatalyst tests and possibly reused.

Reviewer: 2

Comments to the Author(s)

In this manuscript, the authors present the synthesis, structure and its photocatalytic properties of a new HPOTa complex $K_3[H_3P_4(TaO_2)_6(OH)_4O_{20}] \cdot 12H_2O$, which was successfully used to

degrade the Rhodamine B, Methyl blue and Acid red 1. The characterization and data analysis of this manuscript is sufficient. Therefore, the manuscript can be considered to be accepted after further revision. The following questions needs to be addressed for further improvement:

1. The nature of catalyst is that the composition of catalyst does not change before and after the catalytic reaction. In order to prove that the as-obtained material is a catalyst, please provide the phase data of the material after the catalytic reaction.
2. Dye-degradation catalysts have been studied for decades. Many corresponding catalysts have been developed. Please compare the previous reports on dyestuff degradation catalysts and propose the advantages of using HPOTa complex as a catalyst. The author can refer to relevant research, such as *Nanoscale*, 2013, 5, 6589-6598; *Nanoscale*, 2014, 6, 1124-1133; *CrystEngComm*, 2012, 14, 3965-3971; *Chemical Engineering Journal*, 2018, 334, 1537-1549.
3. Please propose the mechanism of catalytic degradation.
4. There are some typo and grammar mistakes. Please revise the manuscript carefully.

Author's Response to Decision Letter for (RSOS-190015.R0)

See Appendix A.

RSOS-190015.R1 (Revision)

Review form: Reviewer 1

Is the manuscript scientifically sound in its present form?

Yes

Are the interpretations and conclusions justified by the results?

Yes

Is the language acceptable?

Yes

Do you have any ethical concerns with this paper?

No

Have you any concerns about statistical analyses in this paper?

No

Recommendation?

Accept with minor revision (please list in comments)

Comments to the Author(s)

In the revised manuscript, my previous concerns are partially addressed. Since this reported dye decolorization by the compounds can be useful in applications other than wastewater treatment, this manuscript can be accepted for publication after minor revision, with the following concerns properly addressed.

1. Figure S17, for the polyoxometalates reported herein, is it more suitable to use HOMO/LUMO or CB/VB in describing the proposed photocatalytic mechanism?
2. Please refrain from using lumped references. For example, in the first three sentences in the Introduction section, the authors used 7 references, but actually, no more than 3 references are needed. There is no need to have 37 papers to support the first two paragraphs of the Introduction. The authors are strongly advised to remove the unnecessary references.

Decision letter (RSOS-190015.R1)

24-Jun-2019

Dear Dr Zhou:

Title: Photocatalytic decolorization of three commercial dyes using a new heteropolyoxotantalate catalyst

Manuscript ID: RSOS-190015.R1

Thank you for submitting the above manuscript to Royal Society Open Science. On behalf of the Editors and the Royal Society of Chemistry, I am pleased to inform you that your manuscript will be accepted for publication in Royal Society Open Science subject to minor revision in accordance with the referee suggestions. Please find the reviewers' comments at the end of this email.

The reviewers and handling editors have recommended publication, but also suggest some minor revisions to your manuscript. Therefore, I invite you to respond to the comments and revise your manuscript.

Because the schedule for publication is very tight, it is a condition of publication that you submit the revised version of your manuscript before 03-Jul-2019. Please note that the revision deadline will expire at 00.00am on this date. If you do not think you will be able to meet this date please let me know immediately.

- 1) A text file of the manuscript (tex, txt, rtf, docx or doc), references, tables (including captions) and figure captions. Do not upload a PDF as your "Main Document".
- 2) A separate electronic file of each figure (EPS or print-quality PDF preferred (either format should be produced directly from original creation package), or original software format)

- 3) Included a 100 word media summary of your paper when requested at submission. Please ensure you have entered correct contact details (email, institution and telephone) in your user account
- 4) Included the raw data to support the claims made in your paper. You can either include your data as electronic supplementary material or upload to a repository and include the relevant doi within your manuscript
- 5) All supplementary materials accompanying an accepted article will be treated as in their final form. Note that the Royal Society will neither edit nor typeset supplementary material and it will be hosted as provided. Please ensure that the supplementary material includes the paper details where possible (authors, article title, journal name).

Best wishes,
Dr Laura Smith
Publishing Editor, Journals
Royal Society of Chemistry
Thomas Graham House
Science Park, Milton Road
Cambridge, CB4 0WF
Royal Society Open Science - Chemistry Editorial Office

RSC Associate Editor:
Comments to the Author:
(There are no comments.)

RSC Subject Editor:
Comments to the Author:
(There are no comments.)

Reviewer comments to Author:
Reviewer: 1

Comments to the Author(s)
In the revised manuscript, my previous concerns are partially addressed. Since this reported dye decolorization by the compounds can be useful in applications other than wastewater treatment,

this manuscript can be accepted for publication after minor revision, with the following concerns properly addressed.

1. Figure S17, for the polyoxometalates reported herein, is it more suitable to use HOMO/LUMO or CB/VB in describing the proposed photocatalytic mechanism?
2. Please refrain from using lumped references. For example, in the first three sentences in the Introduction section, the authors used 7 references, but actually, no more than 3 references are needed. There is no need to have 37 papers to support the first two paragraphs of the Introduction. The authors are strongly advised to remove the unnecessary references.

Author's Response to Decision Letter for (RSOS-190015.R1)

See Appendices B & C.

Decision letter (RSOS-190015.R2)

30-Jul-2019

Dear Dr Zhou:

Title: Photocatalytic decolorization of three commercial dyes using a new heteropolyoxotantalate catalyst

Manuscript ID: RSOS-190015.R2

It is a pleasure to accept your manuscript in its current form for publication in Royal Society Open Science. The chemistry content of Royal Society Open Science is published in collaboration with the Royal Society of Chemistry.

RSC Associate Editor

Comments to the Author:

Thank you for making the minor revisions requested by the reviewer. The manuscript can now be accepted.

Reviewer(s)' Comments to Author:

Appendix A

Dear Dr Laura Smith,

Thank you for your work to our manuscript entitled “**Photocatalytic decolorization of three commercial dyes using a new heteropolyoxotantalate catalyst**” (*Manuscript ID: RSOS-190015*). We have carefully checked and revised our manuscript in accordance with the referee and Subject Editor suggestions. Also, the reference style has been formatted according to this Journal. Revised portions are highlighted in the revised manuscript and the revised supporting information.

In addition, the second unit of the first author, Yansong Wang, is updated from “*Collaborative Innovation Center on Yellow River Civilization Heritage and Modern Civilization Construction, Henan University*” to “*National Demonstration Center for Environmental and Planning, Henan University*” owing to the inner administrative adjustment of our University.

We hope this revised manuscript is acceptable.

Best regards,

Dr. Yansong Wang and Zhimin Zhou

Comments from Reviewer 1:

- This manuscript reported the synthesis of a new heteropolyoxotantalate and tested its performance for photocatalytic degradation of three organic dyes in water. The materials are generally well characterized. However, this reviewer has concerns over the rationale for using this homogeneous photocatalyst for dye removal, since the compound is hard to remove from water after photocatalytic test. Actually, this may be the reason why the compound (and other homogeneous photocatalysts) are more suitable for solar energy applications such as hydrogen generation instead of environmental applications such as water purification. To be accepted for publication, the authors need to address the above concern and perform experiments to show that this compound can be easily recovered from the treated water after photocatalyst tests and possibly reused.

Response: Good point! In our experiment, compound **1a** is almost dissolved in RhB solution, and the recovery rate of compound **1a** is only *ca.* 20%, which is collected by centrifugation. The stability of compound **1a** in solution can be proved by the following three experiments. First, the time-scale UV spectra suggest that the characteristic absorption peak of compound **1a** at *ca.* 250 nm remains unchanged over 12 h (Figure S10a in the revised Supporting Information). Second, it can be clearly seen that the absorption peak of RhB (554 nm) is disappeared after irradiation while the peak at *ca.* 250 nm still exists in the UV spectra for the compound **1a** and Rhb mixed solution (Figure S10b in the revised Supporting Information). Third, the negative-ion ESI-MS spectra indicate that the main envelope observed at $m/z = 597.5$ for **1a** could be assigned to the formula $[H_7\{P_4Ta_6(O_2)_6O_{24}\}]^{3-}$, which can be also remained for 24 h (Figure S11 in the revised Supporting Information). These results ensure the catalytic role of compound **1a** under homogeneous condition.

Moreover, the IR spectrum (KBr pellet) of compound **1a** retains very well before and after catalysis (Figure S12 in the revised Supporting Information), indicating that compound **1a** can be stable in the solid state.

Figure S10. a) The time-scale UV spectra for the solution of compound **1a**; b) the UV spectra of compound **1a** and RhB mixed solution before and after irradiation.

Figure S11. Negative-ion ESI-MS spectra of cluster anion in compound **1a** during 24 h. It can be seen that the main envelope observed at $m/z = 597.5$ for **1a** could be assigned to the formula $[\text{H}_7\{\text{P}_4\text{Ta}_6(\text{O}_2)_6\text{O}_{24}\}]^{3-}$, which can be also remained for 24 h.

Figure S12. IR spectra of compound **1a** before and after catalysis, highlight the region between 1250 to 400 cm^{-1} .

As you mentioned above, photocatalytic water-splitting offers a promising way for environmentally friendly solar-hydrogen production in recent years. Therefore, a preliminary photocatalytic study for hydrogen generation has been done, and the results show that the H_2 evolution rate is $3383 \mu\text{mol h}^{-1} \text{g}^{-1}$ for compound **1a** (100 mg) over 6 h with the corresponding turnover number of 432 (moles of H_2 formed / moles of **1a**), which is shown in Figure S16 in the revised Supporting Information. Also, blank experiments indicate that no H_2 can be detected under the absence of Pt-co-catalyst, sacrificial solvent (CH_3OH) or cluster **1a**.

Figure S16. Time courses of photocatalytic H_2 evolution under the presence of compound **1a** (black) and the absence of compound **1a** (red). Experiment condition: 100 mg of **1a** and 0.05 mg of H_2PtCl_6 were dissolved in 100 mL of 10% methanol, which was irradiated under UV using a 500 W mercury lamp.

Comments from Reviewer 2:

- In this manuscript, the authors present the synthesis, structure and its photocatalytic properties of a new HPOTa complex $K_3[H_3P_4(TaO_2)_6(OH)_4O_{20}] \cdot 12H_2O$, which was successfully used to degrade the Rhodamine B, Methyl blue and Acid red 1. The characterization and data analysis of this manuscript is sufficient. Therefore, the manuscript can be considered to be accepted after further revision. The following questions needs to be addressed for further improvement:

1. The nature of catalyst is that the composition of catalyst does not change before and after the catalytic reaction. In order to prove that the as-obtained material is a catalyst, please provide the phase data of the material after the catalytic reaction.

Response: Thank you for your positive comments and good suggestion. As above, the stability of compound **1a** dissolved in solution can be proved by both UV and ESI-MS spectra, shown in Figures S10 and S11 in the revised Supporting Information. Whereas the IR spectra of compound **1a** remain the same before and after catalysis (Figure S12 in the revised Supporting Information), indicating that compound **1a** can be stable in the solid state.

Figure S10. a) The time-scale UV spectra for the solution of compound **1a**; b) the UV spectra of compound **1a** and RhB mixed solution before and after irradiation.

Figure S11. Negative-ion ESI-MS spectra of cluster anion in compound **1a** during 24 h. It can be seen that the main envelope observed at $m/z = 597.5$ for **1a** could be assigned to the formula $[\text{H}_7\{\text{P}_4\text{Ta}_6(\text{O}_2)_6\text{O}_{24}\}]^{3-}$, which can be also remained for 24 h.

Figure S12. IR spectra of compound **1a** before and after catalysis, highlight the region between 1250 to 400 cm^{-1} .

- 2. Dye-degradation catalysts have been studied for decades. Many corresponding catalysts have been developed. Please compare the previous reports on dyestuff degradation catalysts and propose the advantages of using HPO₂a complex as a catalyst. The author can refer to relevant research, such as *Nanoscale*, 2013, 5, 6589-6598; *Nanoscale*, 2014, 6, 1124-1133; *CrystEngComm*, 2012, 14, 3965-3971; *Chemical Engineering Journal*, 2018, 334, 1537-1549.

Response: Thank you for your good suggestion. POMs is a large family of metal oxo clusters used for various applications as it has a lot of advantages like, structural versatility, accessible redox property, being environmental friendly and chemically stable. A number of investigations on the photocatalytic activity of POM clusters are also studied. As you said, dye-degradation catalysts have been studied for decades. Honestly, we have paid more attention on POMs instead of the other materials in photocatalytic properties.

Thank you for your recommendation! We have carefully read these four researches and cited them in our revised manuscript. However, it is difficult to compare the dye-degradation effect owing to the different experiment conditions, such as dye concentration, hydrogen peroxide concentration, pH, light source (UV light or visible light) and the amount of reported clusters. For example, most experiments were carried out with the help of H₂O₂ to yield highly reactive hydroxyl radicals that can in turn react with the dyes and lead to dye degradation.

In our experiment, the degradation of RhB could be negligible under 350 W Xenon lamp irradiation in the absence of the title compound **1a**, while the degradation rate was greatly enhanced upon introducing compound **1a**. After 270 min irradiation, about 90% of RhB was removed in the presence of the **1a** samples. However, your suggestion gives us an inspiration to design experiment in the future, we should pay much attention to the related researches and do comparison that maybe be more persuasive.

- 3. Please propose the mechanism of catalytic degradation.

Response: Thank you for your good suggestion. As shown in Figure S17 in the revised Supporting Information, the charge separation may occur when the compound absorbed the UV light. The adsorbed oxygen present in the system and then form the active species $O_2^{\cdot -}$ to degrade the RhB. So, we think it might be a one-step mechanism for compound **1a**.

Figure S17. Schematic illustration for the photocatalytic mechanism of compound **1a**.

- 4. There are some typo and grammar mistakes. Please revise the manuscript carefully.

Response: We have modified the English expression and polished the whole manuscript, highlighted in yellow in the revised manuscript. Thank you!

Appendix B

Dear Dr Laura Smith,

Thank you for your work to our manuscript entitled “**Photocatalytic decolorization of three commercial dyes using a new heteropolyoxotantalate catalyst**” (*Manuscript ID: RSOS-190015.R2*).

We have contacted the CSD team in regards to the DOIs for the data records. The response is: CSD team do not assign DOIs to data pre-publication. They only assign DOIs to data once it has been published. The only solution is to publish our data as a CSD Communication, which will make the data accessible to the public. However, this should not be used if we are planning to publish our data in a scientific article.

Then, we got advice from Editorial Coordinator, Lianne Parkhouse and cited the CCDC datasets appropriately within the revised manuscript.

In addition, we have already published the file we deposited to the Royal Society figshare portal.

We hope this revised manuscript is acceptable.

Best regards,

Dr. Yansong Wang and Zhimin Zhou

Appendix C

Dear Dr Laura Smith,

Thank you for your work to our manuscript entitled “**Photocatalytic decolorization of three commercial dyes using a new heteropolyoxotantalate catalyst**” (*Manuscript ID: RSOS-190015*). We have carefully revised our manuscript according to the suggestion from reviewer 1. Revised portion are highlighted in the revised manuscript and the revised supporting information.

We hope this revised manuscript is acceptable.

Best regards,

Dr. Yansong Wang and Zhimin Zhou

Comments from Reviewer 1:

- In the revised manuscript, my previous concerns are partially addressed. Since this reported dye decolorization by the compounds can be useful in applications other than wastewater treatment, this manuscript can be accepted for publication after minor revision, with the following concerns properly addressed.
1. Figure S17, for the polyoxometalates reported herein, is it more suitable to use HOMO/LUMO or CB/VB in describing the proposed photocatalytic mechanism?

Response: Thank you for your positive comments and good suggestion. Figure S17 has been revised in accordance with your suggestion.

Figure S17. Schematic illustration for the photocatalytic mechanism of compound 1a.

- 2. Please refrain from using lumped references. For example, in the first three sentences in the Introduction section, the authors used 7 references, but actually, no more than 3 references are needed. There is no need to have 37 papers to support the first two paragraphs of the Introduction. The authors are strongly advised to remove the unnecessary references.

Response: Thank you for your advice. We generally will cite two or more references to make our point, which leads to some unnecessary references in the introduction section of our manuscript. And, we have only cited the representative references and refrained the lumped references in accordance with your suggestion. Thus, the reference number was decreased from 61 to 39 in the revised manuscript.